# B-Cell Epitope Mapping of the *Vibrio cholera* Toxins A, B, and P and an ELISA Assay

**DOI:** 10.3390/ijms24010531

**Published:** 2022-12-28

**Authors:** Salvatore G. De-Simone, Paloma Napoleão-Pêgo, Priscilla S. Gonçalves, Guilherme C. Lechuga, Sergian V. Cardoso, David W. Provance, Carlos M. Morel, Flavio R. da Silva

**Affiliations:** 1Center for Technological Development in Health (CDTS)/National Institute of Science and Technology for Innovation in Neglected Diseases Populations (INCT-IDPN), Oswaldo Cruz Foundation (FIOCRUZ), Rio de Janeiro 21040-900, RJ, Brazil; 2Epidemiology and Molecular Systematics Laboratory (LEMS), Oswaldo Cruz Institute, Oswaldo Cruz Foundation (FIOCRUZ), Rio de Janeiro 21040-900, RJ, Brazil; 3Program of Post-Graduation on Science and Biotechnology, Molecular and Cellular Biology Department, Biology Institute, Federal Fluminense University, Niterói 24020-036, RJ, Brazil; 4Department of Health, Graduate Program in Translational Biomedicine (BIOTRANS), University of Grande Rio (UNIGRANRIO), Caxias 25071-202, RJ, Brazil

**Keywords:** cholera toxin, B-cell epitope, peptide microarray, synthetic peptide, MAPs, ELISA

## Abstract

Oral immunization with the choleric toxin (CT) elicits a high level of protection against its enterotoxin activities and can control cholera in endemic settings. However, the complete B-cell epitope map of the CT that is responsible for protection remains to be clarified. A library of one-hundred, twenty-two 15-mer peptides covering the entire sequence of the three chains of the CT protein (CTP) was prepared by SPOT synthesis. The immunoreactivity of membrane-bound peptides with sera from mice vaccinated with an oral inactivated vaccine (Schankol™) allowed the mapping of continuous B-cell epitopes, topological studies, multi-antigen peptide (MAP) synthesis, and Enzyme-Linked Immunosorbent Assay (ELISA) development. Eighteen IgG epitopes were identified; eight in the CTA, three in the CTB, and seven in the protein P. Three *V. cholera* specific epitopes, Vc/TxA-3, Vc/TxB-11, and Vc/TxP-16, were synthesized as MAP4 and used to coat ELISA plates in order to screen immunized mouse sera. Sensitivities and specificities of 100% were obtained with the MAP4s of Vc/TxA-3 and Vc/TxB-11. The results revealed a set of peptides whose immunoreactivity reflects the immune response to vaccination. The array of peptide data can be applied to develop improved serological tests in order to detect cholera toxin exposure, as well as next generation vaccines to induce more specific antibodies against the cholera toxin.

## 1. Introduction

The Gram-negative bacterium *Vibrio cholera* causes cholera, a life-threatening diarrheal disease that is spread by ingesting contaminated food or water. This pathogen continues to be of global importance, with epidemics occurring largely in developing countries that lack sufficient infrastructure to treat sewage and provide clean water, such as Southeast Asia, Latin America, and parts of Africa. Worldwide, 1.3–4 million cases of cholera and 20,000–140,000 cholera-related deaths are recorded yearly. Approximately 50% of those affected are children aged five years or younger [1,2]. In addition, climate change and warming temperatures have spread *V. cholerae* to new geographical areas, highlighting the urgent need for further research to improve our understanding of this pathogen [3,4].

The strains of *V. cholerae* that are known to cause disease and pandemics are categorized in the O1 or O139 serogroups. Although most *V. cholerae* are “non-O1/O139” environmental strains, they may or may not cause some form of gastroenteritis [5,6]. The disease is highly contagious and characterized by symptoms such as the classical profuse watery diarrhea, which often leads to extreme dehydration, shock, and eventual death [6]. According to WHO reports, more than 80% of cholera patients can be successfully treated via oral rehydration therapy [7].

The complete toxin is a hexamer with an 83 kDa protein that belongs to the AB class of bacterial toxins, denoted by AB5. The single heterodimeric holotoxin A subunit (CTA) has a molecular weight of 28 kDa and is responsible for the toxicity of the bacteria, whereas the homopentameric B (five identical 11.0 kDa polypeptides) subunit (CTB5) has a molecular weight of 55 kDa. It binds the holotoxin to the eukaryotic cell receptor in order to transport the bacteria into the cell. The A subunit has a specific enzymatic function that acts intracellularly [8]. The intracellular target of CT is adenyl cyclase, which is one of the most important regulatory systems in eukaryotic cells. This enzyme mediates the transformation of ATP into cyclic AMP (cAMP), a crucial intracellular messenger in various cellular processes. In addition, CT catalyzes the transfer of ADP-ribose from NAD (Nicotinamide Adenine Dinucleotide) to a specific arginine residue in the target (Gsa) protein [8].

Consequently, this results in the activation of adenyl cyclase and, therefore, an increase in the intracellular level of cAMP which activates cAMP-dependent protein kinase. This leads to protein phosphorylation, altered ion transport, and diarrhea with massive fluid loss [9]. Therefore, public health interventions are critical in order to limit the dissemination of cholera, which can otherwise easily become epidemic, particularly in areas without proper sanitation and clean water.

Immunization with cholera vaccines is an important strategy in order to control and prevent cholera epidemics or pandemics. *V. cholerae* virulence factors, including CT [10,11], toxin-coregulated pili [12], lipopolysaccharide (LPS), [13] and outer membrane proteins (Omps) [14,15,16], are ideal candidates for designing a cholera vaccine. Today, two types of oral vaccines are available: attenuated oral cholera vaccines (aOCVs) and killed oral vaccines (kOCVs). The aOCVs consist of attenuated, whole *V. cholerae* cells that display direct efficacy in protecting populations in endemic regions [17,18]. These vaccines were recently deployed during outbreaks in non-endemic areas as part of “reactive” vaccination programs in order to block the spread of cholera [19,20]. kOCVs are effective in inducing heard protection for at least three years and are effective tools for cholera control. In addition, one dose provides short-term protection at least, which has important implications for outbreak management. However, the optimal efficacy of kOCVs requires two doses administered 14 days apart, and the vaccine needs refrigeration [21]. These features may limit the capacity of kOCVs to rapidly constrain ongoing outbreaks in destabilized or resource-limited settings. Nevertheless, single-dose live aOCVs have shown satisfactory efficacy in challenge studies [22] and early-phase clinical trials in endemic regions [23]. The design of reactive vaccination programs with a live aOCV may have the best chance of decreasing the incidence of cholera during outbreaks [24,25].

Oral cholera vaccination is a rapid approach to preventing outbreaks in at-risk settings and controlling cholera in endemic settings [26,27]. However, the benefits of vaccine-derived immunity may be short-lived due to human mobility combined with waning vaccine efficacy. As the supply and utilization of oral cholera vaccines expand, critical questions must be addressed concerning the coverage and vaccine application schedule necessary to generate herd immunity [28]. Serological diagnostic assays have the potential to be employed for large population studies [29] to provide information on the duration of the immune response, which could be improved by the identification of epitopes in CT that are recognized by the antibodies produced by a vaccine. This would advance our understandings of response continuance and herd protection.

An efficient, high-content screening method for epitope mapping involves screening peptide libraries that represent target coding sequences. By synthesizing peptides directly onto cellulose membranes, it is possible to generate refined maps of the complete epitome in viruses and the hundreds of epitopes possible in a large protein [30,31]. Commonly referred to as SPOT synthesis analysis, this strategy was applied here in order to identify the linear B-cell epitopes in three toxins from *V. cholerae*: enterotoxins A (8), B (3), and P (7). A total of 18 linear B-cell epitopes were identified. Three were chosen to perform peptide ELISAs and evaluated using sera from mice that were vaccinated with oral cholera. Two specifically reactive sequences were observed, further validating the systematic definition of pathogen epitomes.

## 2. Results

### 2.1. Identification of the Immunodominant IgG Epitopes in Cholera Toxin

Epitopes in the three components of CT (CTA (258aa), CTB (124aa), and CTP (221aa)) were identified based on the recognition of representative peptides. These were synthesized as a library and detected using mice antibodies immunized with an oral *V. cholera* vaccine (15 days post vaccination; see Section 4). The data in Figure 1A,B present the position of each peptide and the measured intensity, respectively, from the chemiluminescent detection of mouse IgG antibodies present in sera pooled from mice vaccinated with aOCV. The overlapping of the spots with signal strengths above 30% defined the minimum sequence of the epitopes. A list of the synthetic peptides and their positions on the membrane is presented in Appendix A. The sequences constituting the reactive peptides defined 18 IgG epitopes of sizes ranging from 4 to 14 aa that were generated by vaccination with aOCV (Table 1).

### 2.2. Localization of the Major B Epitopes within the Three Enterotoxins

The 18 linear B epitopes identified by the SPOT synthesis analysis were distributed throughout Vc/TxA, Vc/TxB, and Vc/TxP (Figure 2). The enterotoxin chain A1 contains six epitopes (Vc/TxA-1 to Vc/TxA-6), chain A2 houses two epitopes (Vc/TxA-7 and Vc/TxA-8), enterotoxin B has three (Vc/TxB-9 to Vc/TxB-11), and seven epitopes are present in enterotoxin P (Vc/TxP-12 to Vc/TxP-18) (Table 1).

**Table 1 ijms-24-00531-t001:** List of B-cell linear IgG epitopes identified in TxA, TxB, and P deduced from the overlap of consecutive positive peptides with signal intensities greater than 30%.

Protein Code	aa	Sequence	2nd Structure *	Code	Peptide Search **
P01555	26–30	ADSRP	C	Vc/TxA-1/miG	Various bacteria
	37–45	SGGLMPRGQ	C	Vc/TxA-2/miG	Sp
	66–77	TQTGFVRHDDGY	C	Vc/TxA-3/miG	Sp
	108–115	TAPNMFNV	C	Vc/TxA-4/miG	*E. coli*
	176–180	AADGY	C	Vc/TxA-5/miG	Various bacteria
	191–204	AWREEPWIHHAPPG	C	Vc/TxA-6/miG	Sp
	244–245	YQSD	C	Vc/TxA-7/miG	Various bacteria
	250–258	THNRIKDEL	C	Vc/TxA-8/miG	Sp
P01556	20–25	HGTDQN	C	Vc/TxB-9/miG	Sp
	53–56	AGKR	C	Vc/TxB-10/miG	Sp
	64–77	KNGAIFQVEVPGSQ	C + S + C	Vc/TxB-11/miG	Sp
P29485	17–21	NECTN	C	Vc/TxP-12/miG	Various bacteria, virus
	26–40	AQDPMKPERLIGTPS	C + S + C	Vc/TxP-13/miG	Sp
	53–59	YHPAPCP	C	Vc/TxP-14/miG	Sp
	68–77	WPHGFISSESL	C + H	Vc/TxP-15/miG	Sp
	90–95	NDEHKT	C	Vc/TxP-16/miG	Sp
	112–121	VIVSENVVDE	S + C	Vc/TxP-17/miG	Sp
	174–177	GITH	C	Vc/TxP-18/miG	Various bacteria

Sp, specific epitopes; C, coil; H, helix; S, strand; * based on an I-TASSER analysis; ** UNIPROT.

### 2.3. Spatial Location of the Reactive Epitopes of Enterotoxin A, B, and P

The eighteen linear B epitopes identified by the SPOT synthesis analysis were distributed throughout the *V. cholera* toxin proteins (Figure 2). The *V. cholera* protein gene contains three well-defined segments: a short signal peptide in the N-terminal extension (aa 1–18), the A1 subunit chain segment (aa 19–212), the A2 subunit chain segment (aa 213–258), the B subunit (aa 1–124), and the P protein (aa 1–221).

The structures of the enterotoxin A, B, and P proteins were obtained from the protein data bank (PDB) or were predicted using AlphaFold DB to access the location of the epitopes. Alternatively, the tertiary structure was deduced using the I-TASSER server (http://zhanggroup.org/I-TASSER/, accessed on 20 March 2022). The predicted structural models of the enterotoxin A, B, and P proteins were obtained (Figure 2A–C) and displayed the possible spatial localization of the eight reactive epitopes in CTA, the three epitopes in CTB, and the seven epitopes in CTP. Most of the identified epitopes were in loop/coil structures, which were present on the protein surface and accessible to the solvent (Figure 2).

### 2.4. Specific and Cross-Immune IgG Epitopes

To investigate possible cross-immunity conferred by the CT proteins, the sequences of CTA, CTB, and CTP were used as templates in a multi-peptide match that searched for peptides deposited in the UniProtKB data bank. The search criteria were for four or more consecutive amino acids that were identical. This analysis suggested that twelve epitopes from *V. cholerae* are expected to be specific (Table 1), whereas five (Vc/TxA-1, Vc/TxA-5, Vc/TxA-7, Vc/TxP-12, Vc/TxP-18) were expected to display cross-reactivity with various bacteria. The *V. cholera* specific epitopes in CTA were Vc/TxA-2, Vc/TxA-3, Vc/TxA-6, and Vc/TxA-8). In CTB, the specific epitopes were Vc/TxB-9, Vc/TxB-10, and Vc/TxB-11. The epitopes Vc/TxP-13, Vc/TxP-14, Vc/TxP-15, Vc/TxP-16, and Vc/TxP-17 were determined to be specific in CTP (Table 1).

### 2.5. Epitope Reactivity by ELISA

The potential for serological asays based on bacterial proteins to display cross-reactivity is a well-known limitation and was a major impetus for identifying the individual linear B-cell epitopes in the cholerae enterotoxins involved in the vaccination process. We reasoned that by restricting the antigens used in an assay to capture reactive antibodies to those definitively specific to V. cholorea by sequence analysis, a diagnostic ELISA could be developed that eliminated the potential for false positive results. To that end, three epitopes were chosen: one from the major antigenic region of chain A (TQTGFVRHDDGYGGG; Vc/TxA-3), one from chain B (KNGAIFQVEVPGSQG; Vc/TxB-11), and one from the protein P (WPHGFISSESLGGGG; Vc/TxP-15). These epitopes were chosen to generate multi-antigen peptides (MAPs) by solid-phase synthesis using the F-moc strategy. The schematic structures of the dendrimers are shown in Figure 3.

The sera from each immunized mouse reacted with all three synthetic peptides that were derived from *V. cholera* without reacting to the negative control peptide QEVRKYFCV (*Vaccinia virus;* Figure 1A G9, G17). Importantly, no reactivity was observed with the sera from non-vaccinated healthy animals. These results are shown in Figure 4A and were calculated to be significant (*p* < 0.001).

Based on a Receiver Operating Characteristics (ROC) curve analysis (Figure 4B), the area under the curve (AUC) for the epitope Vc/TxB-11 was 0.9987 (*p* < 0.0001), as detected by ELISA with a confidence interval of 95%, demonstrating high diagnostic accuracy for the peptide. The Vc/TxB-A3 peptide presented a reactivity lower than the peptide Vc/TxB-11 epitope, but both demonstrated 100% sensitivity and specificity. The values of the ROC curve were determined as the reactivity to Vc/TxP-15 peptide was much weaker.

## 3. Discussion

Vaccine production is usually a laborious and costly experimental process. Nevertheless, these immunobiological molecules are the most potent countermeasures used to prevent infectious diseases. Today, two different types of cholera vaccines exist that are commercially available or under development: inactivated vaccines, such as Dukoral^®^ (Crucell, Netherlands, 1990, denatured vCh-), mORCAX™ (VaBiotech, Hoan Kiem District, Hanoi, Vietnam) and Sanchol™ (Shantha, Biotechnics-Sanofi Pasteur, Muppiriddipalli, Telangana, India), Euvichol or EuvicholPlus (EuBiologics Co., Seoul, Republic of Soul Korea), and Dukoral (Valneva, Kirkland, QC, Canada), and live attenuated vaccines, such as CVD103-HgR and Peru-15 or CholeraGarde and Vaxchora (Emergent BioSolutions, Rockville, MD, USA) [29,32,33]. Although it has been proven that both types are protective for cholera, the immunological mechanisms dependent on B-cell and/or T-cell immunity are unknown [29]. It is recognized that the most important stimulators of innate immunity and the subsequent adaptive immune response are the LPS O antigen and CT [6,10], which activate the NF-κB and IL-1 systems which are critical factors for promoting long-term mucosal protection [8,13,34,35]. Thus, perhaps the most important question remains how to create more efficacious cholera vaccines, as current vaccine efficacies are only ~60% [27,36]. Therefore, the identification of immunodominant epitopes that interact with the antibodies generated in response to vaccination can contribute to the understanding and selection of better antigens.

In this work, we experimentally identified all the linear B epitopes in the CT recognized by IgG antibodies from mice vaccinated with the inactivated Sanchol™ vaccine using a peptide microarray analysis. CT is the primary virulence factor of *V. cholerae*, and ingestion of as little as 5 g produces the symptoms of cholera [37]. The reactivity of the serum from vaccinated animals against a subset of regions in CT showed that the vaccine has a high antigenic potential with only a single dose. Eight B linear epitopes were readily identified in CTA, three in CTB, and seven in protein CTP. All eighteen epitopes were exposed on the molecular surface and were accessible to the immune system (Figure 3).

The CTA subunit is synthesized as a single polypeptide with a signal peptide (aa 1–18). Enzymatic activity is initiated after proteolytic cleavage, “nicking” at residue Arg-192, which produces two functional domains: A1 (~21.8 kDa, aa 19–204) and A2 (~5.4 kDa, aa 206–258). The two fragments remain covalently associated by a disulfide bond. The B subunit contains five chains that form a pentameric ring around a central pore structure. CTA1 is responsible for ADP-ribosyltransferase activity, and the alpha-helical CTA2 has an α–helix tail that moors the CTA1 and CTB5 subunits together [38,39,40,41]. CTB is a non-toxic subunit that produces target cells that block antibodies and is a potent mucosal immunostimulatory adjuvant that can incite potent mucosal and immune system responses [42]. Because of the strong adjuvanticity of CTB, it is applied in different peptide and DNA vaccines, such as microbial and cancer vaccines [43,44]. In addition to playing a role as an immunostimulatory adjuvant, CTB is also a powerful protective antigen.

CTA and CTB seem to have the greatest protective importance of the three chains that constitute the cholera toxin. Initially, CTA was not considered to have immunological importance, despite having the toxic portion of the molecule [34]. However, this assumption has been revised due to recent studies that showed that anti-CTA antibodies are capable of toxin neutralization at very low concentrations in persons living in areas where cholera is endemic [9]. This ability has been attributed to the cross-reactivity of CTA epitopes with the enterotoxigenic *Escherichia coli* (ETEC) heat-labile toxin (LT) [45] together with the frequent exposure of people living in endemic areas to ETEC LT toxin [46]. Our study mapped three epitopes in CTA that present an extensive potential to cross-react with bacterial toxins (Table 1), more so than the other subunits.

Studies describing the identification of CT epitopes are rare in the literature. Nevertheless, previous cross-reaction studies using polyclonal rabbit sera and B-toxin synthetic peptides neutralization have identified the presence of three immunogenic peptides, aa 8–20, aa 50–64 (anti-CP3) [47], and aa 69–85, in the CTB [48]. However, our studies also identified similar immunogenic peptides using sera from mice that were orally vaccinated (Table 1). They were identified as Vc/TxB-9miG (aa 20–25), Vc/TxB-10/miG (aa 53–56), and Vc/TxB-11/miG (aa 64–77). In another study in mice, it was demonstrated that the anti-CTB specific immune response was dominant and protective [49]. Yet, circulating levels of CTB-IgG antibodies and CTB-specific memory B-cells wane within months after natural infection [50,51], as well as in human challenge studies [42,52]. Together, CTB provides only a short-term immunity boost [49]. More concerningly, overall protection with the CTB-containing vaccine appeared to drop to lower levels than the non-CTB-containing vaccine within two years after vaccination [42]. Therefore, the three epitopes (Vc/TxB-9/miG, Vc/TxB-10/miG, and Vc/TxB-11/miG) identified in the B subunit should be of interest for vaccine development despite descriptions against lasting protection conferred by the B subunit. Concerning CTP, no detailed studies on its immunogenic importance were found. However, six of the seven identified epitopes are unique and can potentially be used in developing chimeric polyproteins for diagnostic purposes.

Concerning serological diagnoses, the ELISA results using the MAP4-epitope peptides revealed that two of the peptide (Vc/TxA-3 and Vc/TxB-11) epitopes appropriately discriminate between negative and positive samples (*p* < 0.0001). The peptide TxP-15 presented a low immune response 15 days after vaccination; however, their response index increased 30 days post vaccination (Figure 4). The weak immune response induced by the entire peptide P has been characterized [53]. Although they reflect the mouse immune response, these results are important since these epitopes are eligible for phase IIA studies, which involves an estimate of their accuracy (sensitivity and specificity) for the index test in discriminating between diseased and non-diseased people in a clinically relevant population [54]. The performance of the selected epitopes strongly supports the continued use of these epitopes within a chimeric multi-epitope protein [55] as a target in more sensitive and fast diagnostic tests [56].

## 4. Materials and Methods

### 4.1. Immunization of Mice

Thirty Balb C mice (15–21 g) were immunized orally with 20 µL of whole attenuated Schancol^TM^ cholera vaccine (lot SCN021A15) produced by Shantha Biotechnics Ltd. (Muppiriddipalli, Telangana, India). Fifteen days after vaccination, blood samples were obtained, followed by a full bleed at 30 days. For the unvaccinated controls, blood samples were collected from 30 healthy mice. Each group of serum was collected separately, distributed in aliquots of 0.5 mL in Eppendorf tubes, and stored at −20 °C.

### 4.2. Synthesis of the Cellulose Membrane-Bound Peptide Array

A library of one hundred, twenty-two peptides that spanned the entire coding sequence of cholera enterotoxin A (P01555, 258 aa), enterotoxin B (P01556, 124 aa), and Toxin P (P29485, 221 aa) of serotype O1 was synthesized onto a cellulose membrane according to standard SPOT synthesis protocols using an Auto-Spot Robot ASP-222 (Intavis Bioanalytical Instruments AG, Köln, Germany) [57]. Each peptide contained 15 amino acids of coding sequence that overlapped the previous and next peptide by 10 residues. To optimize the library, a GSGSG spacer sequence was inserted at the amino and carboxy termini of each protein (points A1, D5, D6, C4, C5, and F2). Library construction and program execution were conducted with the MultiPep program (Intavis). The peptide QEVRKYFCV (*Vaccinia virus*, spots G9 and G17) was included as a negative control, and the peptides KEVPALTAVETGATN (*Poliovirus*, spots G3 and G11), GYPKDGN AFNNLDRI (*Clostridium tetani,* spots G5 and G13), and YDYDVP DYA GYPYDV (hemagglutinin of *Influenza virus*, spots G7, G15, and G24) were included as positive controls.

Coupling reactions were followed a blocking step by acetylation with acetic anhydride (4%, *v*/*v*) in N, N-dimethylformamide. Next, the fluorenylmethyloxycarbonyl (F-moc) protecting group was removed from the N-terminus of the peptide by adding a 20% solution of piperidine in DMF. This cycle of coupling, blocking, and deprotection was repeated in order to add each successive amino acid until the desired peptide was generated. After adding the last amino acid, the side chains of the amino acids were deprotected using a solution of dichloromethane-trifluoroacetic acid-triisopropylsilane (1:1:0.05, *v/v/v*) and washed with ethanol as described previously [58]. Membranes containing the synthetic peptides were probed immediately.

### 4.3. Screening of SPOT Membranes

The SPOT membranes were washed for 10 min with TBS-T (50 mM Tris, 136 mM NaCl, 2 mM KCl, and 0.05% Tween^®^ 20, pH 7.4) and then blocked with TBS-T containing 1.5% BSA for 90 min at 8 °C under agitation. After extensive washing with TBS-T, the membranes were incubated for 12 h with a pool (n = 10) of vaccinated mouse sera diluted (1:150 for IgG detection) in TBS-T with 0.75% BSA and then washed again with TBS-T. After this, the membranes were incubated with sheep anti-mouse IgG alkaline phosphatase (diluted 1:5000), prepared in TBS-T with 0.75% BSA for 1 h, and then washed with TBS-T and CBS (50 mM citrate-buffered saline). Finally, the chemiluminescent substrate Nitro-Block II was added to complete the reaction. The integrity of the library synthesis was confirmed with the reactivity of the control peptides to human sera.

### 4.4. Scanning and Measurement of Spot Signal Intensities

Chemiluminescent signals were detected on an Odyssey FC (LI-COR Bioscience, Lincoln, NE, USA) using the same conditions described previously [59] with minor modifications. Briefly, a digital image file was generated at a resolution of 5 MP, and the signal intensities were quantified using the TotalLab TL100 (v 2009, Nonlinear Dynamics, Newcastle-Upon-Tyne, UK) software using its automated grid search for 384 spots. The signal intensities (SI) were exported to Excel (Microsoft Corp., Redmond, WA, USA) in order to remove background signals (signal at the negative control) and normalized to a percent of the highest signal measured. An epitope was defined in the sequences of two or three positive contiguous spots that presented a normalized SI greater than or equal to 30% [60]. For three or more contiguous spots ≥ 30%, epitopes were identified visually when three or more contiguous spots were positive.

### 4.5. Preparation of the Multi-Antigen Peptides

For the preparation of the dendrimer multi-antigen peptides (MAPs), a solid phase synthesis protocol was performed with tetrameric (F-moc) 4-Lys2-Lys-β-Ala Wang resin (CEM Corp, Charlotte, NC, USA) as previously described [59]. The peptides possessed the sequence: (A) Vc/TxA-3 (TQTGFV RHDDGYGGG), (B) Vc/TxB-11 (KNGAIFQVEVP GSQG), and (C) Vc/TxP-15 (WPHGFISSESLGGGG). The constructs were prepared in an automated peptide synthesizer (MultiPep-1, CEM Corp, Charlotte, NC, USA), and the side chains of tetrafunctional F-moc amino acids were protected with TFA-labile protecting groups as required. Once sequence assembly was completed, the F-moc groups were removed, and the peptide resin was cleaved and fully deprotected with TFA/H2O/EDT/TIS (94/2.5/2.5/1.0 *v*/*v*, 90 min). The peptides were precipitated by adding chilled diethyl ether, centrifuged (30,000× *g*, 10 min at 4 °C), and the pellet was dissolved in aqueous AcOH (10% *v*/*v*), dried, and stored as a lyophilized powder. When necessary, the MAP was dissolved in water, centrifuged (10,000× *g*, 60 min at 15 °C), and the supernatant was filtered by a Centricon™ (Merck Millipore, Burlington, MA, USA) ten filter. Purification of the MAPs was performed at BIOMANGUINHOS-FIOCRUZ, Rio de Janeiro, Brazil. An XBridge BEH C18 (2.7 μ, 5 cm × 4.6 mm) column coupled to a water autopurify HPLC system (Water Corporation, Newcastle, NS, Australia) at a flow rate of 1.2 mL min^−1^ using mobile phases A [0.05% formic acid in water (18 MΩ × cm)] and B [(0.05% formic acid in ACN (acetonitrile acid)] (*v/v*) in water and a gradient of 0–97 B in 40 min. Detection at 200–300 nm using a diode array was used during purification.

For ESI-TOF, the peptides were solubilized in deionized water to a final concentration of 10 µg/mL and then formic acid was added to a final concentration of 0.1%. The mass spectrometer used was the Water UPLC model Acquity-I Class (Water Corp., Newcastle, Australia), and the samples were injected at 1 µL/min. The range used for ion detection was from 1000 to 11,500 *m*/*z*.

### 4.6. ELISA

Peptide-based ELISA assays were performed as described previously [60]. Briefly, wells of 4HB NUNC plates for ELISAs were coated with 50 ng of each peptide in 100 µL of coating buffer (Na_2_CO_3_–NaHCO_3_, pH 9.6) overnight at 4 °C. Next, the plates were washed three times using PBS-T washing buffer (PBS with 0.1% Tween^®^ 20, pH 7.2) and blocked using PBS-T with 2.5% BSA over 2 h at 37 °C. Next, diluted mouse sera (100 µL) in coating buffer was applied and incubated for 2 h at 37 °C. Following several washes with PBS-T, the plates were incubated for 2 h at 37 °C with 100 µL of goat anti-mouse IgG conjugated to alkaline phosphatase (Sigma, St Louis, MO, USA) diluted in coating buffer, washed, and exposed to PNPP substrate (Sigma, St Louis, MO, USA). Absorbance was measured at 405 nm on a FlexStation 3 Microplate Reader (Molecular Devices, Sunnyvale, CA, USA). Initially, the optimal serum dilution value of 1:100 was determined by a titration series of vaccinated mouse sera and by constructing the receiver operating characteristic (ROC) curve.

### 4.7. Structural Localization of the IgG Epitopes and Bioinformatics Tools

To ascertain the location of the epitope within the 3D molecular structure of enterotoxin A, enterotoxin B, and toxin coregulated pilus biosynthesis protein P (Toxin P) from *V. cholerae*, we constructed in silico protein models using the I-TASSER server (http://zhanglab.ccmb, accessed on 25 March 2022). Models were chosen according to best C-score and TM-score (topological evaluation value) [61]. The resulting 3D structural models were also analyzed in the AlphaFold database [62]. Data bank searches for *V. cholerae* sequence homologies were performed using previously identified sequences and annotated proteins in other organisms in the database UniProt (http://www.uniprot.org/, accessed on 20 April 2022) and multiple peptide match (https://research.bioinformatics.udel.edu/peptide match/index.jsp, accessed on 10 May 2022).

### 4.8. Statistical Analysis

Data were analyzed using the program R (version 3.6.0) and R Studio. The paired *t*-test was applied in order to compare the statistical significance between the two samples. Statistical significance was considered at *p* ≤ 0.05. GraphPad Prism version 5.0 was used to analyze the receiver operating characteristic curve (ROC).

## 5. Conclusions

A high-throughput immune response profile using a peptide array that was directly generated on a cellulose membrane facilitated the identification of the major antigenic determinants in enterotoxins A, B, and P, which were recognized by antibodies from mice that were orally vaccinated with a single dose of the aOCV. Eighteen IgG epitopes that were distributed throughout the bacterial CT proteins were identified. Six epitopes were identified in the A1 chain of the enterotoxin, two in the A2 chain, three in enterotoxin B, and seven in enterotoxin P. Six of the epitopes showed a level of similarity to the proteins in other pathogens that suggested a high potential for cross-reactivity. Twelve *V. cholera* specific epitopes were defined, and their spatial localizations within protein structural models were revealed using bioinformatics. The performance of three of these epitopes (Vc/TxA-3, Vc/TxB-11, and Vc/TxP-15) to detect antibodies generated in mice as a response to vaccination was empirically confirmed by an indirect ELISA. The molecular characterization of linear B-cell IgG epitopes in cholera enterotoxins can be used to construct a polyepitope chimeric protein for use in next generation rapid diagnostic tests and highlights the utility of epitope mapping to better understand the immune response to current and future vaccines in order to improve the production of specific neutralizing antibodies.

## Figures and Tables

**Figure 1 ijms-24-00531-f001:**
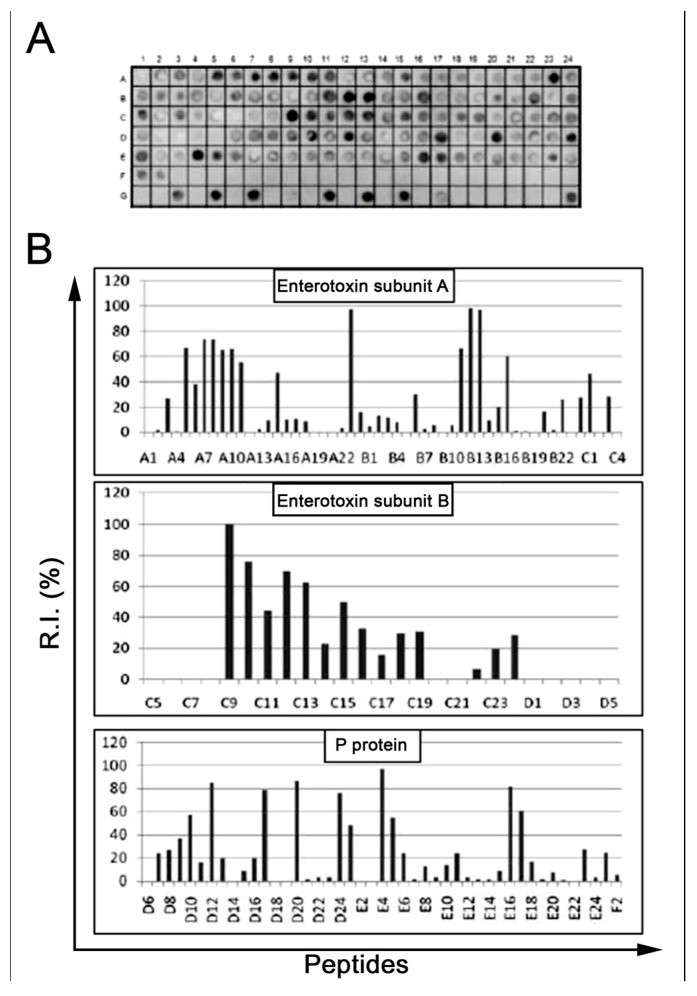
Fine epitope mapping of *Vibrio cholerae* toxin A (P01555), B (P01556), and P (P29485). A membrane-bound peptide library representing the three chains of the toxin was probed with a pool of vaccinated mouse sera (n = 15), and reactivity was detected using goat anti-mouse IgG alkaline phosphatase alkaline labeled secondary antibody and chemiluminescence substrate. Panel (**A**) presents an image of the peptide array showing reactivity as dark circles. The panels in (**B**) graphically present the percent signal after normalizing the signals to the positive and negative controls. The sequences of peptides in each position are listed in Appendix A.

**Figure 2 ijms-24-00531-f002:**
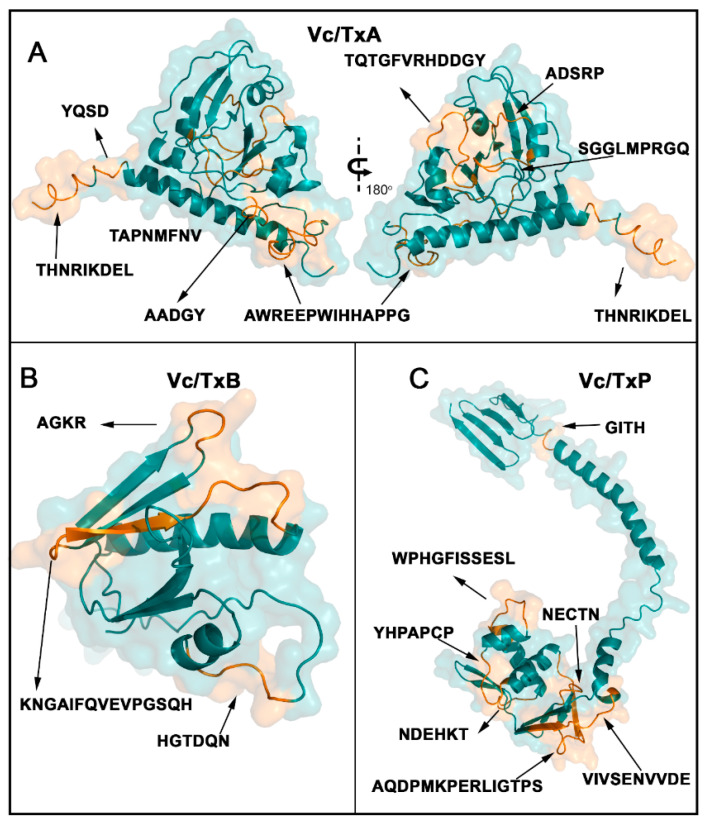
Epitope localization in three-dimensional structures of *V. cholerae* toxins A (**A**), B (**B**), and P (**C**). Epitopes are presented in orange within models constructed using the crystal structure of toxin A (PDB: 1xtc) and the AlphaFoldDB predicted protein structures of toxins B and P. Images were created using PyMOL.

**Figure 3 ijms-24-00531-f003:**
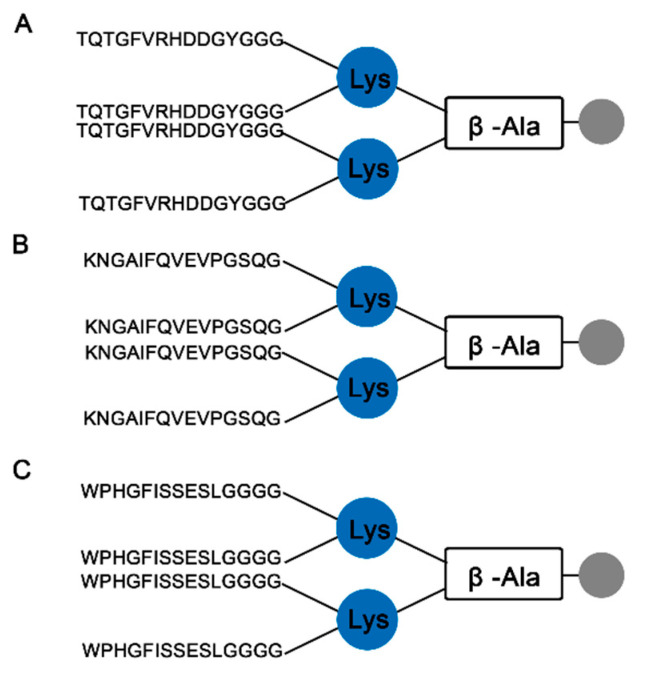
Structure of the three MAP4s synthesized for peptide ELISAs. The Wang resin used for F-moc peptide synthesis generated a dendrimer with four copies of the epitope (**A**) TxA-3, (**B**) TxB-11, and (**C**) TxP-15.

**Figure 4 ijms-24-00531-f004:**
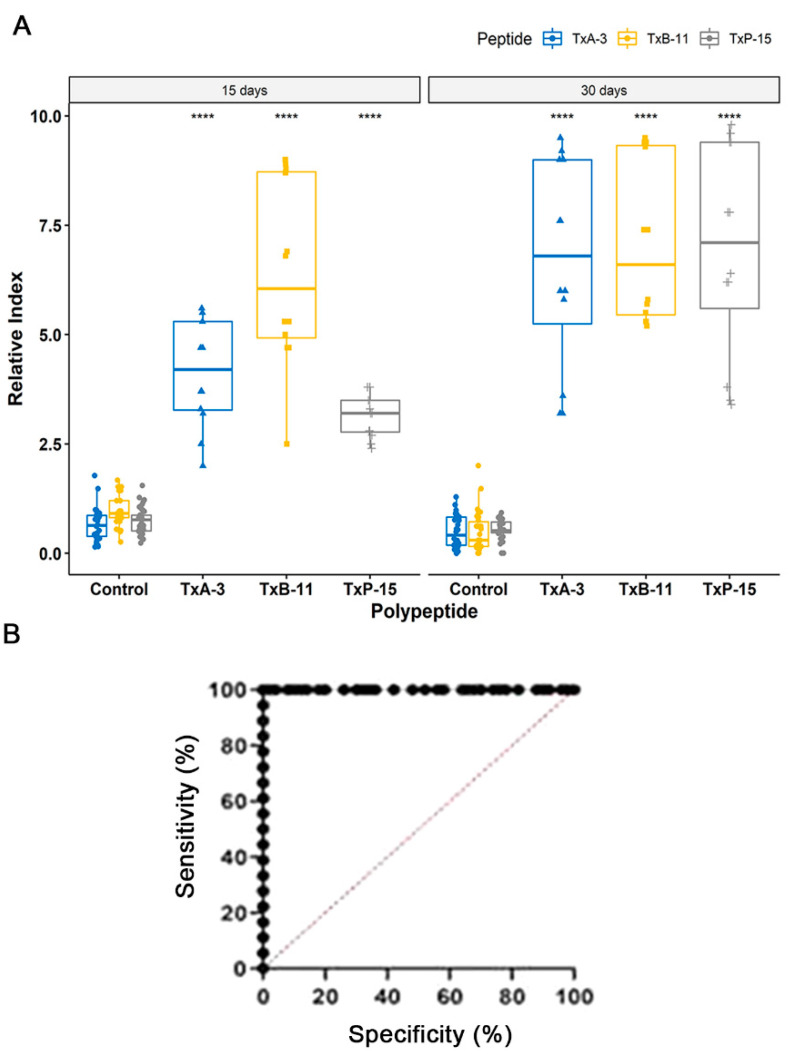
Reactivity of vaccinated mice sera (n = 12) 15- and 30-days post vaccination against the MAP4 peptides Vc/TxA-3, Vc/TxB-11, and Vc/TxP-15 by ELISA. (**A**) Unvaccinated mice were used as a control (n = 30). The relative index was calculated using the cut-off of each MAP to normalize the measured value from the sera of 12 vaccinated mice. (**B**) The receiver operating characteristic curve (ROC) was used to determine the cut-off of peptide B (Vc/TxB-11; 0.053), reactivity (100%), specificity (100%), and AUC (0.9987). (****) Statistical significance at *p* < 0.0001.

## Data Availability

The data presented in this study are available on request from the corresponding author.

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
