# Peer review of "B-Cell Epitope Mapping of the Vibrio cholera Toxins A, B, and P and an ELISA Assay"

_ijms, 2022, doi:10.3390/ijms24010531_

Round 1
Reviewer 1 Report (Previous Reviewer 1)
The manuscript was improved in several aspects, however multiple flaws still persist.
In materials and methods, it is stated that 30 non-vaccinated mice were evaluated and then in the results, it was referred to pre-vaccinated mice. It is not the same to use a non-vaccinated control group comparing it with vaccinated groups as to evaluate the same mice before and after vaccination. This must be clarified.
The graphs of the reactivity of the negative sera should also be shown and show the statistical differences with the sera of the vaccinated animals. How to evaluate the specificity without taking into account the values of the negative sera? That was not very clear to me in the manuscript.
In materials and methods, there should be a section describing the statistical methods used
Author Response
1) In materials and methods, it is stated that 30 non-vaccinated mice were evaluated and then in the results, it was referred to pre-vaccinated mice. It is not the same to use a non-vaccinated control group comparing it with vaccinated groups as to evaluate the same mice before and after vaccination. This must be clarified.
A: Thanks for the note. The word has been modified to unvaccinated.
2) The graphs of the reactivity of the negative sera should also be shown and show the statistical differences with the sera of the vaccinated animals. How to evaluate the specificity without taking into account the values of the negative sera? That was not very clear to me in the manuscript.
R: Thank you, figure 4 has been modified and the column corresponding to negative sera has been inserted. Regarding specificity and sensitivity, the values are provided when using the ROC curve, which already includes the values of negative sera. No other statistical methods were used to calculate these values.
3) In materials and methods, there should be a section describing the statistical methods used.
A: Thanks for the note, we really don't understand why this sub item was not inserted in the text. Now we are entering all statistical methods used in the work
Reviewer 2 Report (Previous Reviewer 3)
While appreciating the efforts of the authors to meet the requirements for publication, still my professional ethics prevent me to accept the article in its present form, that is without full characterization of the new compounds, which is still missing.
Supplementary figure 2, which is unreadable, is the only characterization reported by the authors. Its quality must be improved. The gradient used must be reported.
Mass spectra of the new compounds are still missing, but clearly they must be included in any research articles where synthesis is reported.
To me the paper cannot be published without full characterizations of the new compounds.
Actually, I am unable to recommend acceptance of any papers reporting synthesis without characterizations of the compounds synthesized.
Author Response
1) While appreciating the efforts of the authors to meet the requirements for publication, still my professional ethics prevent me from accepting the article in its present form, that is without full characterization of the new compounds, which is still missing.
R: Thanks again for the referee's commitment and the observations made, which were undoubtedly necessary to improve the quality of the work.
2) Supplementary figure 2, which is unreadable, is the only characterization reported by the authors. Its quality must be improved. The gradient used must be reported.
R: A new figure with the larger characters has been introduced and the HPLC gradient values described in the legend.
3) Mass spectra of the new compounds are still missing, but clearly they must be included in any research articles where synthesis is reported.
A: A new supplementary figure has been introduced showing the mass spectrum of the TxB-11 major functional peptide. We did not find it necessary to attach the graphs for the TxA-3 and TxP-15 peptides, as the results for these peptides were unsatisfactory.
Round 2
Reviewer 1 Report (Previous Reviewer 1)
The questions and observations appear to have been addressed by the authors, but the submitted version of the manuscript does not appear to be the one updated with the corrections. Please verify this.
Author Response
Thanks for your precious time and effort in giving valuable comments and suggestions to our manuscript.
Reviewer 2 Report (Previous Reviewer 3)
While sure that the authors did report it as claimed, I'm afraid I was not able to find the gradient, I'm sorry about that. I found: "An XBridge 351
BEH C18 (2.7 μ, 5 cm x 4.6 mm) column coupled to an HPLC system (Water Autopurification System) at a flow rate of 1.2 ml min-1 using mobile phases A [0.05% formic acid in water (18 MΩ × cm)] and B [(0.05% formic acid in ACN (acetonitrile)] (v/v) in water." both in the materials and methods and in the legend to Figure S2 of the manuscript uploaded for review. The gradient should be something like "10-100%B in 40 min". I must say it is quite worrying that the absorption is as low as 2 AU (usually it is around 200).
Figure s3. The mass spectrum should have "m/z" on the x-axys; I assume "MASSAS-electrospray" means a ESI-TOF was used, but it should be stated. The instrument used should also be reported (both the name of the instrument, e.g., Xevo or Micromass, and the brand, e.g., Waters or Agilent). The MS spectrum reported (Figure s3) seems a deconvoluted spectrum after processing rather than the real spectrum. If so, it should be stated in the legend to Figure S3.
Author Response
Referee 2
While sure that the authors did report it as claimed, I'm afraid I was not able to find the gradient, I'm sorry about that. I found: "An XBridge 351BEH C18 (2.7 μ, 5 cm x 4.6 mm) column coupled to an HPLC system (Water Autopurification System) at a flow rate of 1.2 ml min-1 using mobile phases A [0.05% formic acid in water (18 MΩ × cm)] and B [(0.05% formic acid in ACN (acetonitrile)] (v/v) in water." both in the materials and methods and in the legend to Figure S2 of the manuscript uploaded for review. The gradient should be something like "10-100% B in 40 min". It is quite worrying that the absorption is as low as 2 AU (usually around 200).
R: Thanks for your precious time and effort in giving valuable comments and suggestions to our manuscript. The gradient has been completed (0-97 % B), and we agree that the analytical information from the purification and analysis of the peptides should be explicit. We apologize for the omissions, but as reported in the work, these analyzes were carried out in other laboratories, and unfortunately, we were not given all the information they should have been. We hope that now, with the new information inserted, we have overcome all obstacles. Regarding the UA, the analysis performed using the (U) detector (HPLC) has sufficient sensitivity, focused on the ultraviolet (range 190-350 nm), to assess the purity of the peptide, which is the objective of this work.
Figure s3. The mass spectrum should have "m/z" on the x-axis; I assume "MASSAS-electrospray" means an ESI-TOF was used, but it should be stated. The instrument used should also be reported (both the name of the instrument, e.g., Xevo or Micromass and the brand, e.g., Waters or Agilent). The MS spectrum reported (Figure s3) has a deconvoluted range after processing rather than the real spectrum. If so, it should be stated in the legend in Figure S3.
R: Thank you; these were corrected in the legend of FigS3. The x and y-axis were updated to “Relative Absorbance” and “m/z,” using an ESI-TOF and deconvoluted after processing.
This manuscript is a resubmission of an earlier submission. The following is a list of the peer review reports and author responses from that submission.
Round 1
Reviewer 1 Report
The manuscript entitled "Fine epitope mapping of the Vibrio cholera toxins A, B, and P an ELISA assay" has several major flaws
1) The title does not offer an idea of the essence of the work.
Besides, although peptides can be recognized by antibodies as linear epitopes, when speaking of "Fine epitope mapping" T epitopes should be included and this was not done in the study.
2) When trying to find the objective of the work in the abstract and the introduction, apparently the objective was to identify epitopes for vaccine purposes, but then it seems that the objective is the location of epitopes for an ELISA for the diagnosis of cholera. This is not well explained in the work. At times the authors refer to the potential use of that peptides as a vaccine or for diagnosis. That is very confusing. Furthermore, how can a vaccinated person be distinguished from an infected person with these peptides?
3) If the objective is to use these epitopes for diagnostic purposes, it is not through the use of immunization models in mice that their usefulness is determined. For this, sera from people with a confirmed positive diagnosis and the respective negative controls must be used to validate this method. Infection is not the same as vaccination, mainly if mice are being vaccinated and the objective is to diagnose cholera in humans.
4) In Materials and Methods (line 280 - 280) the authors describe that " Fifteen days after vaccination, the serum of 15 animals was obtained a full bleed was followed at 30 days". However, it is not clear if the results shown in Figure 2 correspond to sera from day 15 or 30.
5) The authors concluded that "oral vaccine is effective in inducing long-lasting specific neutralizing antibodies" (line 396 in conclusions), but antibodies of 15 or 30 days after an immunization should not be considered "long-lasting". Why was a curve not made at 0, 15 and 30 days to see the evolution of the antibody titer?.
6) How is it that the authors propose these peptides to be evaluated in phase IIB studies (line 274) without having carried out the previous phases in humans?
In summary, this work is so preliminary. The conclusions are not sufficiently supported by the results, so it needs to be extensively rewritten and additional studies are needed.
Author Response
Thank you very much for reviewing our manuscript and offering valuable advice. We have addressed each of your comments point-by-point below and revised the manuscript accordingly.
1) The title does not offer an idea of the essence of the work. Besides, although peptides can be recognized by antibodies as linear epitopes, when speaking of "Fine epitope mapping" T epitopes should be included and this was not done in the study.
A: We respectfully disagree with the need to include T-cell epitopes for a “fine epitope determination”. In this case, it refers to the resolution of epitope sequences to the minimum sequence (length) of epitopes. We clearly state the objective of the work to map only linear B epitopes through a high resolving peptide microarray technique. It is very complicated to determine T epitopes by this methodology. Normally, this type of epitope is identified through bioinformatics methodologies that use specific algorithms and usually select “greater” peptide sequences than the “minimum” sequence that expresses the minimum number of amino acids of the antigen capable of binding and/or inducing the immune response. Therefore, the title has been maintained to express the original idea of mapping linear B epitopes to the finest resolution.
2) When trying to find the objective of the work in the abstract and the introduction, apparently the objective was to identify epitopes for vaccine purposes, but then it seems that the objective is the location of epitopes for an ELISA for the diagnosis of cholera. This is not well explained in the work. At times the authors refer to the potential use of that peptides as a vaccine or for diagnosis. That is very confusing. Furthermore, how can a vaccinated person be distinguished from an infected person with these peptides?
A: We agree that the phrasing could be improved. Please see lines 25-29
3) If the objective is to use these epitopes for diagnostic purposes, it is not through the use of immunization models in mice that their usefulness is determined. For this, sera from people with a confirmed positive diagnosis and the respective negative controls must be used to validate this method. Infection is not the same as vaccination, especially if mice are being vaccinated and the objective is to diagnose cholera in humans.
A: We studied the immune response in mice because these animals are study models for the cholera vaccine. However, we also understand that the immune response to this same vaccine in humans could identify different peptide sequences. We do not and cannot use sera from patients, because in Brazil, there is no cholera and therefore human antibodies are not accessible to us.
4) In Materials and Methods (line 280 - 280) the authors describe that "Fifteen days after vaccination, the serum of 15 animals was obtained a full bleed was followed at 30 days". However, it is not clear if the results shown in Figure 2 correspond to sera from day 15 or 30.
A: Thank you, the results in Figure 2 were obtained with sera from animals 15 days post-vaccination. As the order of the paragraph was changed, the information was now inserted in the legend of the new figure 4. Please see lines 195-196.
5) The authors concluded that "oral vaccine is effective in inducing long-lasting specific neutralizing antibodies" (line 396 in conclusions), but antibodies of 15 or 30 days after an immunization should not be considered "long-lasting". Why was a curve not made at 0, 15 and 30 days to see the evolution of the antibody titer?.
A: We are in accordance and “long lasting” was removed. To further improve the presentation of the results, the data from the 30-day ELISA was included. Please see line 396 and new figure 4.
6) How is it that the authors propose these peptides to be evaluated in phase IIB studies (line 274) without having carried out the previous phases in humans?
A: It should be written as phase IIA. The informed phase IIB is not a vaccine evaluation (clinical trial), but a diagnostic one, which comprises studies designed to estimate the accuracy (sensitivity and specificity) of the index test in discriminating between diseased and non-diseased people in a clinically relevant population. Please see [Hepatology. 2014, 60, 408-18. doi: 10.1002/hep.26948]. This information has been corrected in the text. Please see lines 269-272.
7) In summary, this work is so preliminary. The conclusions are not sufficiently supported by the results, so it needs to be extensively rewritten and additional studies are needed.
A: Several paragraphs have been rewritten to make the manuscript more understandable. As re-informed in the modified Abstract, the objective was to describe the possible linear B epitopes involved in the cholera toxin protection mechanism.
Reviewer 2 Report
Please see attached.

Author Response
1) For example, this is unclear that how data obtained from Figure 1 lead to the selection of some peptides that were evaluated in Figure 2 and in other experiments.
A: Thank you, we tried to clarify this issue better. Please see item 2.1, lines 109-117.
2) If the peptides were synthesized, provide data on mass and purity of peptides obtained through MS and HPLC
A: This same response was stressed to the referee 3. Our CEM equipment calculates the coupling efficiency in each cycle and also provides the final coupling degree. In the case of the synthesized peptides, it was greater than 95%, indicating a high yield and purity of the peptides. Regardless, it is customary for our laboratory to process the peptides for purification. Our laboratory has been operating as a peptide production facility for the entire Institution for over 20 years and the peptide purification process is performed in the analytical center of Biomanguinhos, another technical unit of Fiocruz. As the graphs of these purifications were above 95% with a confirmational MASS spectra, the laboratory only reported the results but did not preserve the data. Therefore, we rewrote the sentence and informed her performer's name but did not include the purification charts, as we believe it does not contribute any important information to the discussion of the work.
3) The figures provided in the MS are poor quality and lack of axis labels (at least for figure 1)
A: A new figure with higher resolution (600 dpi) was introduced
4) Section 2.3, reference is missing.
A: Sorry, but there is no reference to this data, they are our results.
5) In general, the study is interesting but the results are not well detailed that leads to misunderstanding during reading and results analysis by the readers.
A: When carrying out a new review, we identified that some errors regarding the number and sequence of the analyzed peptides were wrongly inserted. We apologize for the fact, and luckily it was fixed in time.
Reviewer 3 Report
The manuscript by De-Simone and coworkers deals with the synthesis of dendrimers made up of short peptides identified in the epitopes of cholera toxin and the evaluation of their ability to be recognized by antibodies in vaccinated mice serum. The text is well-written and the topic is of interest. Some issues prevent publication of the work in its present form.
1) The authors should clarify (i) if they produced one or more dendrimers and (ii) the peptide sequences involved. They state:
- Abstract, line 24: "a multiple antigen peptide"
- Par. 2.4, line 159: "multi-antigen peptides"
- Discussion, line 265: "MAP4 bi-epitope peptides" and, line 268: "bi-epitope MAP design". By the way, the dendrimer was so far made up of peptides coming from three epitopes, why is it called "bi-epitope"?
The chemical structure of the dendrimer(s) produced MUST be included in the manuscript, as a Figure.
Please, doublecheck the two following statements, which seem in contrast:
Lines 156-160 "To that end, three epitopes were chosen, one from the major antigenic region of chain A1 (aa 37-45; [SGGLMPRGQ] Vc/TxA-2 epitope), one from the chain B (aa 20-25 [HGTDQN]; Vc/TxB-9), and one from the protein P (aa 68-77 [WPHGFISSESL]; Vc/TxP-15), to generate multi-antigen peptides (MAPs) by solid-phase synthesis using the F-moc strategy"
Lines 334-336 "The peptides possessed the sequence: A) Vc/TxA-3 (GGTQTGFVRHDDGYG), (B) Vc/TxB-11 (GKNGAIFQVEVPGSQ), (C) Vc/TxP-13 (GGGGNDEHKTGGGGG)." By the way, the latter sequence does not correspond to that reported in Table 1 for the same compound name (Vc/TxP-13: AQDPMKPERLIGTPS).
2) Synthesis procedures for new compounds should be accompanied by yields and the most important product characterization data. Both yield and characterizations are missing for the MAP(s). If the present work is the first one reporting the synthesis of the particular multiple antigen peptide(s) tested, then the supporting information must include compound characterizations, such as the related HPLC profile and high-resolution (HR)MS spectrum. Otherwise, a reference to the paper where the compound was first described - and characterizations can be found - should be added to the manuscript. In Par. 4.5, lines 349-350, the authors mention 95% purity and MS analysis for identification: the corresponding HPLC and HRMS must be included in the Supporting Information. Please, add the yield to that paragraph.
3) The criteria used to pick the peptide sequences reported in Table 1 should be better explained. Sequences in the Supporting Information are all 15-mers. Materials and Methods, line 286, states: "a library of 15-aa peptides", and line 287, "a ten residue overlap". The authors should add a few statements to explain their experimental procedure, focusing on how they obtain the peptide sequences listed in Table 1, which have different lengths. The reasons to introduce a spacer GSGSG at each subunit terminus (lines 290-291) should also be explained.
Both the Introduction and Discussion sections can be more concise. The discussion on Cholera disease and symptoms can be shortened. The discussion on available vaccines and Cholera virus and disease should be updated taking into account the following, recent review:
Suman Kanungo, et al., Lancet 2022; 399: 1429–40. https://doi.org/10.1016/S0140-6736(22)00330-0
Minor points
Abstract. Lines 18-19, "Here, we ... Spot synthesis": please, remove or rephrase the statement, since it apparently doesn't make sense. Did the authors map the epitopes prepared by Spot synthesis?
Introduction. Line 64, "molecular weight of 85.3 kDa": please, doublecheck the value. Lines 75-78, "Therefore, ... acquired [17]": "Therefore"? The statement is out of context, please move it somewhere else. Line 88, "heard".
Results. Line 123, "human IgG"? Lines 124-125, "Intensities were normalized...(data not shown". The statement is too generic. Data normalization must be explained at least in the related Material&Methods paragraph. Legend to Fig. 1, line 133, "labeled secondary and chemiluminescence substrate": statement not clear; lines 134-135, "Panels B graphIcally presenT ...based on normalizing ...controls": statement not clear, please rephrase/complete - or remove it. Line 147-148, typo "(Error! ... valid.)". Line 160, "Fmoc" (not F-moc). Line 166, "pre-vaccination animals?": the reported number of mice and the Materials&Method section do not include sera from "pre-vaccinated" animals. Please, clarify. Line 167, "ROC": please, explain abbreviations when first encountered in the text. Legend to Fig. 2, line 174, "sera from non-vaccinated mouse sera"? Paragraphs. 2.5 and 2.3 should be moved together to improve readability.
Discussion. Line 205, "the immunological mechanisms ... are unknown": the statement is too generic and not completely true, please rephrase it. Lines 231-232, "is a potent mucosal .... that can incite potent mucosal ... responses": please, rephrase the statement. Lines 237-238, "Initially, ... molecule.": please add a reference. Line 239, "recent studies", please add the corresponding references. Line 253, "The observation supports this hypothesis"? Statement not clear. Line 264, "developing safer vaccines": please, explain on what ground would existing vaccines be less safe or remove the last few words of the statements.
Materials and Methods. Line 286, "and Toxin coregulated a library"? Statement not clear. Line 298, "to render the peptides N-reactive during subsequent steps": acetylation caps unreacted free amines! Please, correct or remove the statement. Lines 299-300, "the F-moc protecting groups were removed by adding piperidine to make the nascent peptides reactive": please, rephrase the statement. The fluorenylmethyl protecting group was removed from the peptide N-terminus by adding a 20% piperidine solution in DMF. The step is a deprotection of the amine at the N-terminus of the peptide, not an activation. Please, add the coupling procedure. Par. 4.4, lines 324-326, "To be considered an epitope, ... signal intensity (SI) greater than or equal to 30% of the highest value obtained from the set of spots on the respective membrane". Please, add a reference in support to this method: "the highest value" of the test is not objective, perhaps a positive control was used? Par. 4.5, line 333, "tetrameric Fmoc4-Lys2-Lys-B-Gly Wang resin", perhaps super- and subscripts were lost? Please use standard chemical expressions (e.g, B, Boc?). Line 346, "flow rate of 1.2 ml/min": please doublecheck the flow rate used for purification.
Conclusions. Lines 393-395, "The molecular characterization .... perspective": Too generic. Please, either offer examples of the potential benefits for "both applied and basic research" or remove the statement. Lines 395-397, "and confirms the oral vaccine is effective ... antibodies": The manuscript does not report a study on the induction of long-lasting specific neutralizing antibodies by the identified compounds. Please, remove the statement.
Author Response
Thank you very much for reviewing our manuscript and offering valuable advice. We have addressed your comments with point-by-point responses and revised the manuscript accordingly.
1) The authors should clarify (i) if they produced one or more dendrimers and (ii) the peptide sequences involved. They state: - Abstract, line 24: "a multiple antigen peptide" - Par. 2.4, line 159: "multi-antigen peptides"
A: Thank you, the term has been standardized throughout the text. Regarding the number of peptides and the sequence, this seems to be explicit in the Abstract (line 24) and section 2.4 (line 159). However, we agree that it would be clearer to insert the complete sequence. Please see line figure 3 and lines 336-338..
2)- Discussion, line 265: "MAP4 bi-epitope peptides" and line 268: "bi-epitope MAP design." By the way, the dendrimer was made up of peptides coming from three epitopes. So why is it called "bi-epitope"?
A: Thank you for your careful review. The context of the sentence was modified during editing in relation to peptides that contained two epitopes (bi-epitope). The phrase has been modified.
3) The chemical structure of the dendrimer(s) produced MUST be included in the manuscript as a Figure.
A: We agree. Please see the new figure 3.
4) Please double-check the two following statements, which seem in contrast:
Lines 156-160 "To that end, three epitopes were chosen, one from the major antigenic region of chain A1 (aa 37-45; [SGGLMPRGQ] Vc/TxA-2 epitope), one from the chain B (aa 20-25 [HGTDQN]; Vc/TxB-9), and one from the protein P (aa 68-77 [WPHGFISESL]; Vc/TxP-15), to generate multi-antigen peptides (MAPs) by solid-phase synthesis using the F -moc strategy."
Lines 334-336 "The peptides possessed the sequence: A) Vc/TxA-3 (GGTQTGFVRHDDGYG), (B) Vc/TxB-11 (GKNGAIFQVEVPGSQ), (C) Vc/TxP-13 (GGGGNDEHKTGGGGG)." The latter sequence does not correspond to that reported in Table 1 for the same compound name (Vc/TxP-13: AQDPMKPERLIGTPS).
A: Thank you, there was an epitope numbering error that has now been fixed in the two sentences.
5) Synthesis procedures for new compounds should be accompanied by yields and the most important product characterization data. Both yield and characterizations are missing for the MAP(s). If the present work is the first one reporting the synthesis of the particular multiple antigen peptide(s) tested, then the supporting information must include compound characterizations, such as the related HPLC profile and high-resolution (HR)MS spectrum. Otherwise, a reference to the paper where the compound was first described - and characterizations can be found - should be added to the manuscript. In Par. 4.5, lines 349-350, the authors mention 95% purity and MS analysis for identification: the corresponding HPLC and HRMS must be included in the Supporting Information. Please add the yield to that paragraph.
A: Our CEM equipment calculates the coupling efficiency in each cycle and also provides the final coupling degree. In the case of the synthesized peptides, it was greater than 95%, indicating a high yield and purity of the peptides. Regardless, it is customary for our laboratory to process the peptides for purification. Our laboratory has been operating as a peptide production facility for the entire Institution for over 20 years and peptide purification process is performed in the analytical center of Biomanguinhos, another technical unit of Fiocruz. As the graphs of these purifications were above 95% with a confirmational MASS spectra, the laboratory only reported the results but did not preserve the data. Therefore, we rewrote the sentence and informed the performer's name but did not include the purification charts, as we believe it does not contribute any important information to the discussion of the work.
6) The criteria for picking the peptide sequences reported in Table 1 should be better explained. Sequences in the Supporting Information are all 15-mers. Materials and Methods, line 286, states: "a library of 15-aa peptides", and line 287, "a ten residue overlap." The authors should add a few statements to explain their experimental procedure, focusing on how they obtain the peptide sequences listed in Table 1, which have different lengths. The reasons for introducing a spacer GSGSG at each subunit terminus (lines 290-291) should also be explained.
A: The phrase has been modified in M&Ms and the legend of Table 1. For brevity, we do not repeat the information in Supporting Information that shows all the synthesized peptides but not the resulting epitopes. Likewise, we do not think it is important to inform how the list of peptides was obtained in the Table since this is disclosed in lines 289-295.
7) Both the Introduction and Discussion sections can be more concise. The discussion on Cholera disease and symptoms can be shortened. The discussion on available vaccines and Cholera virus and disease should be updated, taking into account the following recent review:
Suman Kanungo, et al., Lancet 2022; 399: 1429–40. https://doi.org/10.1016/S0140-6736(22)00330-0
A: The discussion has been edited and includes the suggested reference. Please see 204-209.
Minor points
8) Abstract. Lines 18-19, "Here, we ... Spot synthesis": please, remove or rephrase the statement since it doesn't make sense. Did the authors map the epitopes prepared by Spot synthesis?
A: The Abstract has been rewritten
9) Introduction. Line 64, "molecular weight of 85.3 kDa": please, double-check the value. Lines 75-78, "Therefore, ... acquired[17]": "Therefore"? The statement is out of context. Please move it somewhere else. Line 88, "heard."
A: Thank you, both have been fixed.
Results.
10) Line 123, "human IgG"?
A: Thank you, this has been corrected to read “mouse”.
11) Lines 124-125, "Intensities were normalized..(data not shown". The statement is too generic. Data normalization must be explained at least in the related Material&Methods paragraph. –
A: We have included our reference to the meaning of signal strength [58].
12)-Legend to Fig. 1, line 133, "labeled secondary and chemiluminescence substrate": statement not clear;
A: We agree and have rewritten the sentence.
13)-lines 134-135, "Panels B graphIcally presenT ...based on normalizing ...controls": statement not clear, please rephrase/complete - or remove it.
A: We agree and have rewritten the sentence.
14)-Line 147-148, typo "(Error! ... valid.)". Line 160, "Fmoc" (not F-moc). Line 166, "pre-vaccination animals?": the reported number of mice and the Materials&Method section do not include sera from "pre-vaccinated" animals. Please clarify.
A: Lines 147-148 (typos), Line 160 (Fmoc) and Line 166 (“pre-vaccination) have been fixed.
15)-Line 167, "ROC": please, explain abbreviations when first encountered in the text. Legend to Fig. 2, line 174, "sera from non-vaccinated mouse sera"?
A: The information has been included in both the MM and caption.
16)-Paragraphs. 2.5 and 2.3 should be moved together to improve readability.
A: Thank you, we have taken your advice to rewrite a paragraph.
Discussion.
17)-Line 205, "the immunological statement ... are unknown": the statement is too generic and not completely true, please rephrase it.
A: Thank you, it has been redone, and a reference was included [29].
18)-Lines 231-232, "is a potent mucosal .... that can incite potent mucosal ... responses": please, rephrase the statement.
A: Thank you. The sentence was rewritten. Please see 233-234.
19)-Lines 237-238, "Initially,... molecule.": please add a reference. Line 239, "recent studies", please add the corresponding references.
A: Thank you, two references were introduced.
20)2-Line 253, "The observation supports this hypothesis"? Statement not clear.
A: Thank you, the sentence was rewritten.
21)-Line 264, "developing safer vaccines": please explain on what ground existing vaccines would be less safe or remove the last few words of the statements.
A: We agree and have removed the statement.
Materials and Methods.
22) -Line 286, "and Toxin coregulated a library"? Statement not clear. Line 298, "to render the N-reactive peptides during subsequent steps": acetylation caps unreacted free amines! please,
correct or remove the statement.
A: The paragraph was rewritten to make it more clear.
23)-Lines 299-300, "the F-moc protecting groups were removed by adding piperidine to make the nascent peptides reactive": please, rephrase the statement. The fluorenylmethyl protecting group was removed from the N-terminus peptide by adding a 20% piperidine solution in DMF. The step is a deprotection of the amine at the N-terminus of the peptide, not an activation. Please add the coupling procedure.
A: When an amine is unprotected, the activation of the nascent peptides occurs simultaneously, which in my opinion supports that the sentence is correct. This terminology is used routinely.
To improve clarity, we have reworded the paragraph as suggested by the referee.
24)-Par. 4.4, lines 324-326, "To be considered an epitope, ... signal intensity (SI) greater than or equal to 30% of the highest value obtained from the set of spots on the respective membrane". Please, add a reference in support of this method: "the highest value" of the test is not objective. Perhaps used a positive control?
A: This methodology is semi-quantitative, and the cut-off intensity value may vary depending on the technique and equipment used in the analysis. As our equipment has high sensitivity and due to the vast experience of our group in the method, all our published works always used between 20-30% of the signal strength as the cut-off point. The negative control (line 330) can be a random sequence peptide or the intensity of a point where nothing has been synthesized. The positive control is a peptide from some pathogen commonly used in human vaccination. However, in this case, the serum is from an animal (mouse); therefore, a cholera sequence known from the literature was included as a positive control.
25)-Par. 4.5, line 333, "tetrameric Fmoc4-Lys2-Lys-B-Gly Wang resin", perhaps super- and subscripts were lost? Please use standard chemical expressions (e.g., B, Boc?)
A: The synthesis used was Fmoc and not Boc, and there are no super or subscripts. A parenthesis has been inserted.
26)-Line 346, "flow rate of 1.2 ml/min": please double-check the flow rate used for purification.
A: This value was correct.
Conclusions.
27)-Lines 393-395, "The molecular characterization .... perspective": Too generic. Please, either offer examples of the potential benefits for "both applied and basic research" or remove the statement.
A: Thank you, the sentence has been modified. Please see 397-401.
28)-Lines 395-397, "and confirms the oral vaccine is effective ... antibodies": The manuscript does not report a study on the induction of long-lasting specific neutralizing antibodies by the identified compounds. Please remove the statement.
A: Thank you, the statement was removed.
Round 2
Reviewer 1 Report
The authors adequately answered many of the questions and the manuscript was improved.
Even so, I think that the text should be reviewed carefully because there are still confusing parts. For example at the end of the abstract:
"This array of data and the resulting peptides can be useful in other diverse efforts, including developing more specific antibodies against the cholera toxin"
It is supposed to refer to therapeutic monoclonal antibodies or for diagnostic kit purposes. I think that the authors should be more specific because sometimes they start referring to vaccines and it seems that the objective is to identify vaccine epitopes, then they jump to diagnosis and clinical trials and in the abstract they mention the production of antibodies without clarifying and they do not refer to that in the main text.
I suggest the authors better organize the idea and concentrate it on one objective, although later in the discussion reference is made to other potential uses of these epitopes. Because the way it is organized creates a lot of confusion for readers.
- Nor is it necessary for authors to submit a review article for reviewers to show what is a clinical trial of a diagnostic tool. Those phases are well known. The issue is that the immunogenicity of an epitope demonstrated by vaccination in mice is not the same as the immunogenicity demonstrated by a natural infection in humans. Before proceeding to IIA phase other studies should be performed.
- Check all the citations again E.g. reference 54 is from an article by Glenny (1923). "These results are important although they reflect the mouse immune response since these epitopes are eligible for phase IIA studies, which involves an estimate of their accuracy (sensitivity and specificity) of the index test in discriminating between diseased and non-diseased people in a clinically relevant population [54]". Is this correct?
Author Response
The authors adequately answered many of the questions, and the manuscript was improved.
Even so, I think that the text should be reviewed carefully because there are still confusing parts. For example, at the end of the abstract:
A: The manuscript was re-revied. The changes are in blue.
"This array of data and the resulting diverse peptides can be useful in other efforts, including developing more specific antibodies against the cholera toxin."
It is supposed to refer to therapeutic monoclonal antibodies or for diagnostic kit purposes. I think that the authors should be more specific because sometimes they start referring to vaccines, and it seems that the objective is to identify vaccine epitopes, then they jump to diagnosis and clinical trials, and in the abstract, they mention the production of antibodies without clarifying and they do not refer to that in the main text.
A: A restructuring was carried out.
I suggest the authors better organize the idea and concentrate it on one objective, although later in the discussion, reference is made to other potential uses of these epitopes because the way it is organized creates a lot of confusion for readers.
A: A reconstruction was performed.
- Nor is it necessary for authors to submit a review article for reviewers to show what is a clinical trial of a diagnostic tool. Those phases are well known. The issue is that the immunogenicity of an epitope demonstrated by vaccination in mice is not the same as that of an epitope demonstrated by a natural infection in humans. Before proceeding to IIA phase other studies should be performed.
A: Thank you, removed and fixed.
- Check all the citations again E.g. reference 54 is from an article by Glenny (1923). "These results are important although they reflect the mouse immune response since these epitopes are eligible for phase IIA studies, which involves an estimate of their accuracy (sensitivity and specificity) of the index test in discriminating between diseased and non-diseased people in a clinically relevant population [54]". Is this correct?
A: The reference was changed.

Reviewer 2 Report
The authors have made most of the suggested revisions but Figure 1 is still blurred, lacks y-axis and is not of publication quality. Please consider replacing this with better quality figure.
Author Response
The authors have made most of the suggested revisions, but Figure 1 is still blurred, lacks a y-axis, and is not of publication quality. Please consider replacing this with a better-quality figure. The authors adequately answered many of the questions, and the manuscript was improved.
A: Thank you, the –axis was introduced, and a figure with 600 dpi was introduced.
Reviewer 3 Report
The authors addressed all of the issues rised by the reviewers.
If possible, the following, very small amendment would improve readability of the manuscript:
line 345: please, replace "B" with the greek symbol beta (By the new figure 3, very useful by the way, "B" is - quite probably - for betaAla. At first, I thought it was for "Boc". Perhaps the readers will not be puzzled now, but still).
Other remarks:
Authors' reply (point 5) "we believe it does not contribute any important information to the discussion of the work". I am afraid I have to disagree: HPLC and HRMS spectra of new compounds add very much to the discussion. I am sure the authors are fully aware that several, well-known Editors would not publish articles on new molecules without their characterization.
Authors' reply (point 6): I'm afraid I still think that it would be useful to the readers to know why adding a spacer GSGSG at each subunit terminus. Perhaps a very short explanation can be added to the M&M.
Author Response
We greatly appreciate the reviewer's efforts, comments and suggestions. All suggestions were accepted and inserted into the ceiling. Now the manuscript has been carefully reviewed and several changes have been made, which have been recorded by changes tracked in MSWord.